# The Role of Traditional Livelihood Practices and Local Ethnobotanical Knowledge in Mitigating Chestnut Disease and Pest Severity in Turkey

**Jeffrey Wall** [1,*], **Coşkun Köse** [2,*], **Nesibe Köse** [3], **Taner Okan** [4], **Elif Başak Aksoy** [5], **Devra Jarvis** [6,7] **and Shorna Allred** [1]

1   Department of Natural Resources, Cornell University, Ithaca, NY 14850, USA
2   Department of Forest Biology and Wood Protection Technology, Faculty of Forestry,
    Istanbul University-Cerrahpasa, 34473 Bahçeköy-Istanbul, Turkey
3   Department of Forest Botany, Faculty of Forestry, Istanbul University-Cerrahpasa,
    34473 Bahçeköy-Istanbul, Turkey
4   Department of Forestry Economics, Faculty of Forestry, Istanbul University-Cerrahpasa,
    34473 Bahçeköy-Istanbul, Turkey
5   Department of Anthropology, Faculty of Letters, Hacettepe University, 06800 Beytepe, Ankara, Turkey
6   Bioversity International, Via dei Tre Denari 472/a, Maccarese, 00057 Rome, Italy
7   Department of Crop and Soil Sciences, Washington State University, Pullman, WA 99164, USA
*   Correspondence: jrw297@cornell.edu (J.W); ckose@istanbul.edu.tr (C.K.);
    Tel.: +1-607-252-6579 (J.W.); +90-212-338-2400 (C.K.)

**Abstract:** The European chestnut population is enduring multiple compounding exotic pest and disease outbreaks across Turkey. The deeply held value of the chestnut species for the Turkish public is reflected in substantial government conservation programming. Chestnut is predominantly found on state land managed by Turkey's General Directorate of Forestry (GDF), which generally upholds restrictive policies for chestnut-related livelihood practices other than nut collection and beehive placement. Such policies are justified by a government position that human activities and presence is likely to worsen disease dynamics. Conversely, a growing body of research findings testify that small-scale livelihood practices maintain biological diversity and, furthermore, that this traditional maintenance of diversity has been correlated with decreased pathogen pressure within agroecosystems. However, few studies have investigated this phenomenon in the context of agroforestry systems. At a global ecological moment of increasingly pervasive and severe exotic forest pathogen impact, this paper investigates the influence of diverse small-scale livelihood practices and knowledge on chestnut tree health across the highly heterogenous geography of Turkey. We conducted ethnobotanical questionnaires with 96 chestnut-utilizing households, and chestnut tree health evaluations in georeferenced forest areas they identified, throughout Turkey's Black Sea, Marmara, and Aegean regions. Using data from 1500 trees, we characterized the effects of subsequently recorded environmental, physiological, and anthropogenic factors on tree health using multiple correspondence analysis (MCA), multiple factor analysis (MFA), and mixed models. Our results show that the traditional human management of tree physiology and ecology has a significant positive effect on tree health, especially through the acts of grafting and culling as well as the maintenance of diversity. We argue that conceptualizing such livelihood systems as human niche construction and maintenance can help forest management agencies to better understand and conserve valuable landscapes, even in increasingly common periods of severe pathogenic pressure.

**Keywords:** *Castanea sativa*; human niche construction; mixed effects models; multipurpose forestry

---

## 1. Introduction

This paper employs niche construction theory [1–4] to understand the effects of small-scale livelihood practices on chestnut tree (*Castanea sativa* Mill.) health across Turkey under widespread heavy pest and disease pressure. Turkey is within the geographic center of genetic diversity for the species [5,6], meaning that anthropogenic involvement with the species has historically brought utilization as well as significant biological maintenance. The tree exists today in a wide range of geographic and anthropogenic conditions, from naturalized high-precipitation forest landscapes to dryland irrigated orchards. Today, the entire population endures the compounding effects of multiple exotic pathogen outbreaks introduced variously throughout the last century, including ink disease caused by the oomycetes *Phytophthora cambivora* (Petri) Buisman and *Phytophthora cinnamomi* Rands [7], the chestnut blight caused by the fungus *Cryphonectria parasitica* (Murill) Barr [8], and, most recently, the gall wasp, *Dryocosmus kuriphilus,* Yasumatsu [9].

The two major social forces acting on the chestnut population presently are state management and local livelihood practice. State programs for the conservation of chestnut populations are implemented by the General Directorate of Forestry (GDF). While forest management programs are diverse and locally adapted, GDF generally upholds restrictive policies for chestnut-related livelihood practices other than nut collection and beehive placement. Such policies are justified by a pervasive assumption that human use and interaction leads to ecological erosion. In the case of chestnut, this rests on an assumption that traditional management of the tree and its habitat qualify as irregular or untidy use ("düzensiz faydalanmalar") and only worsen disease dynamics [10] (pp. 7, 11, 18, 20, 22). Here, in order to inform the best forest policy approach to mitigate disease and pest impact in the research context and elsewhere, we present the results of our effort to discern the effects of traditional livelihood practices and ethnobotanical knowledge on the health of chestnut trees while accounting for physiological and geographic factors.

*Human Niche Construction, Traditional Ecological Maintenance, and Forest Policy*

The geneticist Richard Lewontin introduced niche construction theory (NCT) to argue against a metaphorical framework which casts the organism as an influence-taker vis-à-vis an external inviolable environment. This is the kernel of the concept of niche construction, whereby organisms are said to engender in their environment characteristics which benefit their survival and those of their offspring [3]. Humans have been described as the "ultimate niche constructors" ([11]: p. 28). Global erosion of environmental quality may surely be evidence of a niche construction capacity gone awry [4,12]. However, the concept of human niche construction encompasses the human environmental maintenance essential to cultural landscape sustainability.

A growing body of literature documenting the traditional anthropogenic maintenance of biological diversity reveals texture to human niche construction. Biological diversity has been understood as an essential feature of environmental and species' health for decades [13,14]. The maintenance of diversity in domesticated species by traditional livelihoods has been repeatedly observed [15–19]. Traditional agricultural communities maintain and employ crop diversity to ensure population health using a number of agronomic forms such as seed mixtures [20,21], polycultures [22,23], and complex parallel and staggered sowing and harvest schedules for various crops and landraces [24–26].

At the level of landscapes, small-scale societies, both past and present, have implemented widespread and periodic disturbances as a deliberate act of livelihood. Across the eastern and western seaboards of today's United States [27–30], in the Mediterranean [31,32], British Columbia, Canada [33], East Asia [34], and Australia [35,36], environments characterized by routine and widespread anthropogenic disturbance, typically fire, maintain diversity. The creation of landscape mosaics, by fire and other means used by small-scale societies, has been shown to increase species diversity via the moderate multiplication of edge space advantageous for certain biological communities [27,37–39].

There is a growing consensus in ecology and pathology that risk of disease is reduced with increasing biological diversity and that this can occur through several mechanisms at different scales.

At the community level, transmission of a pathogen fatal to one species often occurs via other species for whom the pathogen is not fatal, showing that increasing diversity of community assemblages acts to decrease the likelihood of transmission to the most competent hosts [40]. At the level of the individual species, genetic diversity has been correlated with disease resistance. Inbreeding has been demonstrated to cause the mutation of specific loci necessary for disease resistance [41,42]. Furthermore, spread of infection has been correlated with lower genetic difference with individual like-species neighbors [43].

Many principles from this ecological literature have been shown to be at work in agro-ecological systems, primarily at the level of fields and farms. Polycultures have been demonstrated to imbue disease and pest resistance [44–46]. Varietal mixtures, likewise, have been shown to offer protection against disease advance in populations [47–49]. At the scale of the variety or landrace, studies verify a correlation between intraspecific genetic diversity and disease resistance [50–52].

Although a nascent body of evidence suggests that many previously assumed pristine forests are actually culturally contingent [53,54], few studies probe the dynamics of diversity maintenance and disease resistance—so demonstrable in agroecosystems—in forests. Perhaps as a consequence, this complex anthropogenic support for biological diversity is overlooked and, all too often, constrained by forest policy. Considering the global acceleration of pest and disease pressure on trees and forests so clearly documented in this Special Issue, there is a worldwide urgency to get beyond inflexible and poorly substantiated assumptions that local people only compromise forest and tree health, and instead to understand and support the positive role local people can play. Turkey is no exception. In the definitive GDF publication on the topic, the Chestnut Action Plan 2013–2017 (Kestane Eylem Planı), traditional management of chestnut trees and areas is, as already mentioned, branded as irregular or untidy use [10]. Throughout the text, untidy use is routinely equated with diseases and pests, as a force which acts to exacerbate these negative effects and generally compromise the health of trees (p. 7). For example, it is asserted that the collection of nuts within chestnut forests depletes the seedbank (p. 18). Similarly entrenched attitudes within forest governance institutions are regular challenges to decentralized [55], multipurpose [56], and community forestry [57] initiatives.

We conducted our study in geographical areas where the genetic diversity of the chestnut species has been maintained for millennia. We hypothesized that anthropogenic influences, such as livelihood practices and the knowledge that informs them, on tree health across various landscapes would be significantly independent of other factors. Therefore, this study endeavors to identify and understand the role of traditional livelihood practices and local knowledge in the maintenance of a plant population that is (1) ecologically important [58], (2) indispensable for local livelihoods [59,60], (3) historically anthropogenically maintained [61,62], and (4) under threat and targeted for state conservation [10], while at the same time accounting for the ecological and physical geographic factors that are conventionally analyzed for their effect on plant disease severity.

## 2. Materials and Methods

### 2.1. Study Context

In present-day Turkey, *C. sativa* is distributed across the Eastern Black Sea, Western Black Sea, Marmara, and Aegean regions in a wide variety of elevations, precipitation levels, and temperature regimes [10]. In the Eastern Black Sea, *C. sativa* is found in high year-round precipitation, steep topographic, and heavily forested zones above 400 m above sea level (MASL). In the Western Black Sea region, wet summers and dry winters predominate, and the terrain is more supple, with chestnuts present in more managed forest between 20 to 600 MASL. The *C. sativa* population in the arid Mediterranean climate of western Marmara is found predominantly at elevations higher than 600 MASL, whereas in the famously mild and high precipitation environment of eastern Marmara, *C. sativa* is historically present along the southern coast of the Marmara Sea at elevations as low as 20

MASL. Finally, the warm dry Mediterranean climates of the Aegean region host the most substantial chestnut production in Turkey today, almost all above 800 MASL [10].

These geographic variations are accompanied by significant variation in socioeconomic patterns. Much of this variation is only faintly perceived by governance structures because livelihood practices straddle the governance dimensions of forestry and agriculture. While the most engaged governance entity regarding chestnut trees is, as mentioned, the GDF, there is considerable ambiguity regarding the access rights of smallholders as well as the permission to perform interventions such as grafting, prescribed burning, pruning, and coppicing. This is reflected in the radically variable practice of policy on the ground. In the dense forests of the Eastern Black Sea, chestnut is logged heavily along with numerous other hardwood species. GDF oversees the harvest and collection of all materials from state forests. They issue fee-based permissions for the collection of non-timber forest products to local villagers, and meticulously mediate timber harvest. Seasonal production of honey from the chestnut inflorescence vastly outweighs the collection of chestnuts in livelihood importance. In the Western Black Sea as well as western Marmara, chestnuts are collected predominantly after falling naturally to the forest floor and collection amounts to a tertiary livelihood activity, although the use of chestnut timber is of vital importance for household production. In contrast, in the Aegean and eastern Marmara regions, chestnut cultivation is practiced in orchard settings and represents a premier economic activity. The vast majority of trees are grafted with favored cultivars. In the Aegean sites and throughout most of eastern Marmara, nuts are collected in the husk after being knocked out of the tree with a stick just prior to natural ripening.

## 2.2. Fieldwork

Research was conducted during the summers of 2015 and 2016. In all, 10 sites were selected to represent geographic distribution and diversity of habitat for *C. sativa* (Figure 1). Our approach was household-centered. In each site, 8–10 households were selected through a purposive sampling technique [63] assisted by local forestry officers and/or locally elected village representatives. Household participation was entirely anonymous. By drawing on these sources of local social insight, we identified and engaged with households who were particularly well known for their chestnut-related activities. Our study took no stance on the validity, efficacy, or environmental beneficence of the livelihood practices that would be discussed. Ethnobotanical questionnaires focused on eliciting household knowledge of the morphology, diversity, and management of chestnut trees and how these all related to the performance of livelihood. These questionnaires solicited household knowledge of the following four areas: (1) varieties used, (2) estimated number of trees of each variety managed by the household, (3) land use or access types declared—these types were described by participants, compiled, and later categorized, and (4) plant parts used.

The interviewee household showed us a plot where we conducted our disease severity evaluation on each chestnut tree in a 20 × 20 m georeferenced plot, extending 20 m in a random, recorded direction. Understood as the inverse of tree health, disease severity was defined by the proportion of diseased plant tissue with characteristic symptoms of ink disease, chestnut blight, or gall wasp infection. Respectively, these symptoms included root and whole stem death, cankered and/or blighted stem and branches, and drooping dying leaves with galls on the stems. Severity evaluation was adapted from the procedure proposed by Tizado et al. [64]. Each chestnut tree with a diameter larger than 5 cm was given a disease severity score between 0 and 5 at the main stem (A), the lower crown (B), the mid-crown (C), and the high crown (D), where 0 represented no diseased plant tissue, 1 represented <10%, 2 represented 10%–25%, 3 represented 26%–50%, 4 represented 51%–80%, and 5 represented >80%. In addition, the location of each tree within the plot was recorded using meters forward and meters to the left or right. These data allowed for a later calculation of a geo-coordinate for each tree. Each tree was measured for diameter at breast height (DBH), height, and crown width. Each tree was inspected for evidence of several silvicultural procedures. If the tree was grafted or coppiced, or if major limbs had been removed, this information was recorded.

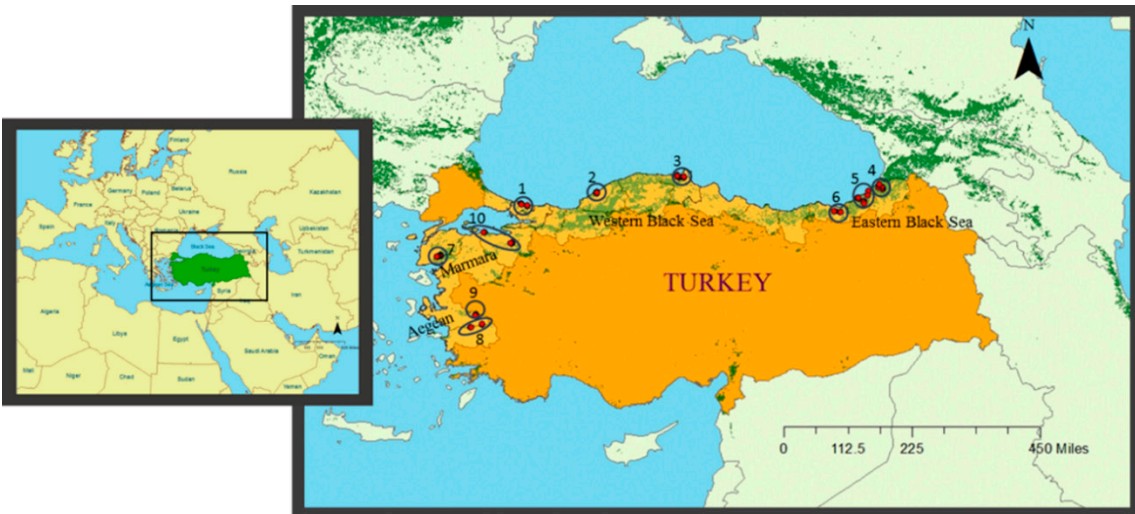

**Figure 1.** Turkey and distribution of hardwood forest in Turkey (derived from Food and Agriculture Organization (FAO) Data in Arc Map). All evaluated sites are indicated with site numbers and province borders. Study sites were located in the following provinces: (1) Şile, (2) Zonguldak, (3) Sinop, (4) Artvin, (5) Rize, (6) Trabzon, (7) Çanakkale, (8) Aydın, (9) İzmir, and (10) Bursa.

## 2.3. Data Analysis

The first phase of data analysis was deriving environmental factors. A series of values for each tree was extracted from secondary datasets in Arc Map software (Version 2.1, Redlands, CA, USA). Elevation, slope, and aspect for each point were derived from a Shuttle Radar Topography Mission (SRTM)-generated digital elevation model (DEM) at 30 × 30 m resolution, housed by the United States Geological Service (USGS) Earth Explorer. We converted aspect into a continuous variable using the heatload value obtained using the Geomorphometry and Gradient Metrics Toolbox (Version 2.0-0, Laramie, WY, USA) [65]. This tool derives a continuous variable which quantifies the relative southwest position of aspect to indicate heat exposure, as south and southwest facing slopes are known to be warmer than north and northeast facing slopes. We extracted the mean of three previous years of normalized difference vegetation index (NDVI) data (2013–2015) for each geocoordinate from National Aeronautics and Space Administration (NASA) Earthdata at 250 × 250 m resolution. The NDVI is a satellite image-derived index indicative of living vegetation presence and of wetness. Finally, we used data for roads and waterways available from Open Street Maps (Version 2018, Cambridge, UK) at 20 × 20 m resolution to derive a "distance to waterway" and "distance to road" value for each tree.

In addition to descriptive statistics, we followed Le Roux et al.'s [66] approach of using multiple correspondence analysis (MCA) to better visualize the variation of disease severity in the surveyed sites. Severity scores for all tree parts (1–5 at A, B, C, and D) of all trees and site number were evaluated as categorical factors in MCA. MCA illustrates relationships between many categorical variables, representing correspondence using proximity in a two-axis plane of constructed dimensions. Multiple factor analysis (MFA) was then used to determine the relations between variable groups—these being (1) environmental continuous variables (DBH, NDVI, distance to water and roads, elevation, slope, aspect), (2) anthropogenic binary factors (grafting, coppicing, trimming), and (3) disease severity as scaled continuous variables—for all parts of all trees. MFA allows for the analysis of groups of variables, including categorical, scaled categorical, and continuous, to be explored linearly and orthogonally to account for potentially correlated, or collinear, factors within and between groups [67–69].

We then sought to characterize the effect of specific factors by using a systematic series of generalized linear mixed models (GLMMs) and linear mixed models (LMMs) using the Akaike information criterion (AIC) score to fit the best model at each stage. GLMM and LMM are an application that allows for the analysis of the effect of multiple variables while accounting for random

effects. GLMM is appropriate when a response variable is irregular or non-normal [70], a common feature of disease severity data in epidemics. It is particularly apt in cases where data exist in subsets which inherently compromise independence. By assigning plot number ($n$ = 97) as a random effect nested in site number ($n$ = 10), our model accounts for the dependence of tree proximity in plots and sites experiencing common disease outbreak dynamics. We began this stage of analysis by generating a binomial variable derived from the mean severity for each tree with "low" being less than 2.5 and "high" being greater than 2.5. We then screened all covariates for correlation with themselves with a step-by-step determination of variable inflation factor (VIF), removing factors with a VIF score above 3 [71]. We analyzed effects on this binomial response variable for all trees ($n$ = 1500) using GLMM.

Each household ($n$ = 96) was associated with a plot with numerous trees as well as specific responses to ethnobotanical questions. Tree-level health data were therefore averaged and the effects of household ethnobotanical knowledge—diversity of cultivars used, diversity of plant parts used, and number of land-use codes exercised—on median plot severity was characterized, using LMM. Linear mixed models are useful for characterizing the relative effect of various factors, while accounting for random effect, when a response variable is normally distributed and continuous [70]. For LMM, site number remained a random effect. After again screening for collinearity, we fit an LMM with household ethnobotanical factors to determine which, if any, of these factors had a significant effect on median plot severity. Finally, with factors determined significant in the first two models, all screened for collinearity, we fit a third LMM to characterize their relative effect on median plot severity, with site number as a random effect. With LMM, we derived a $p$-value by using ANOVA on the model output.

## 3. Results

For this study, 1500 trees on 97 plots in 10 sites were scored for tree health. Overall, 96 households participated in ethnobotanical interviews. In the case of one chestnut plot which we evaluated, the associated household was unable to participate in interview activities. Shown in Figure 2a,b, significant variance in tree health was observed between sites. Three clear tree health groupings of high, medium, and low health are apparent in the MCA output (Figure 2b). We refer to these groups, respectively, as 1, 2, and 3. These can be conveniently remembered as first, second, and third place in tree health. Group 1 includes the sites of Bursa, Aydın, İzmir, and Zonguldak. Group 2 includes the sites of Çanakkale, Sinop, Artvin, Trabzon, and Rize. Group 3 includes the single site of Şile (Figure 1).

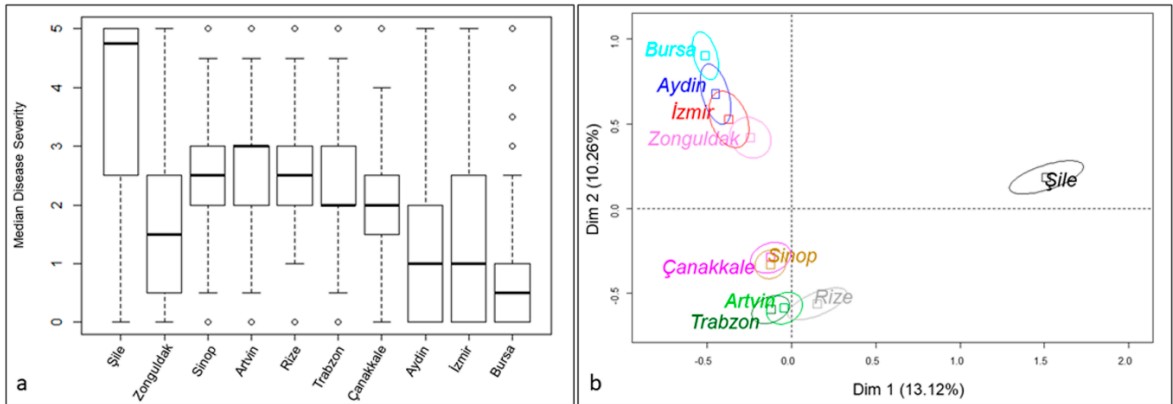

**Figure 2.** (**a**) Boxplot of median disease severity scores of chestnut trees by site number; (**b**) Results of multiple correspondence analysis of all disease severity scores and site number, with site number revealed by a 95% confidence ellipse.

Upon preliminary inspection, these tree health groupings fail to correspond to obvious physical geographic factors. For example, the sites which make up groups 1 and 2 represent no obvious elevational or longitudinal-latitudinal pattern that would arise from natural disease vector or pest spread. Furthermore, the elevation, moisture (NDVI), slope, and tree diameter ranges of groups 1

and 2 overlap considerably (Table 1). Correspondence between tree health and silvicultural context is similarly unconvincing. For instance, while high tree health was observed in areas of intensive, grafted cultivation in İzmir, Aydın, and Bursa, high tree health was also observed in Zonguldak, where grafting was non-existent and collection was small-scale. Medium tree health was observed in the low-intensity collection zones of Çanakkale, Şile, Sinop, Trabzon, Rize, and Artvin. As Table 2 suggests, however, the specific silvicultural practice of grafting is related to low disease severity, a fact which was verified by subsequent analysis.

**Table 1.** Summarizing statistics for recorded factors for blight severity groups as determined by multiple correspondence analysis (MCA).

| Group | Statistical Measures | Diameter (cm) | Elevation (m) | Slope (m) | Wetness (NDVI) | Coppiced (%) | Grafted (%) |
|---|---|---|---|---|---|---|---|
| 3 | Average | 13.8 | 138 | 7.5 | 6016 | | |
|  | Standard deviation | 6.5 | 36.0 | 6.6 | 1541 | 53 | 0 |
|  | Range | 5–33 | 102–222 | 1–19 | 1321–7082 | | |
| 2 | Average | 40.4 | 592 | 18.4 | 6959 | | |
|  | Standard deviation | 37.6 | 253 | 9.2 | 600 | 9 | 2.6 |
|  | Range | 5–320 | 134–1178 | 1–44 | 5392–7950 | | |
| 1 | Average | 37.5 | 551 | 18.8 | 6328 | | |
|  | Standard deviation | 36 | 436 | 6.6 | 966 | 3 | 58 |
|  | Range | 5–241 | 40–1356 | 3–32 | 4497–7503 | | |

**Table 2.** Results of the generalized linear mixed model (GLMM) exploring the effect of all factors on disease severity as a binomial high or low. Factors displayed in order of effect size in Figure 4a. Significance indicated as *** representing 0, ** representing 0.001, * representing 0.01, and • representing 0.05. DF is degrees of freedom. Pr is probability.

| Factor | DF | Chisq | Z-Value | Pr(>|z|) |
|---|---|---|---|---|
| Tree diameter | 1 | 37.8359 | 6.151 | $7.7e^{-10}$ *** |
| Grafted | 1 | 7.7861 | −2.790 | 0.00526 ** |
| Wetness | 1 | 4.3792 | −2.093 | 0.03638 * |
| Slope | 1 | 3.1451 | 1.773 | 0.07615 • |
| Distance to Road | 1 | 1.9748 | −1.405 | 0.15994 |
| Elevation | 1 | 1.3206 | −1.149 | 0.25049 |
| Aspect | 1 | 0.8769 | −0.936 | 0.34905 |
| Limb Removed | 1 | 0.7524 | 0.867 | 0.38572 |
| Coppiced | 1 | 0.1122 | 0.335 | 0.73766 |
| Distance to Water | 1 | 0.0053 | −0.072 | 0.94223 |

Multiple correspondence shows confidence ellipses drawn for each site in the MCA output (Figure 2b). In MCA illustrations, proximity represents correspondence, thus these 95% confidence ellipses represent the centers of correspondence for severity factors. The cluster of sites within the bottom left quadrant are those with the least diseased tissue, whereas the cluster of sites straddling the top right and left quadrant are those of highest diseased tissue. Disease severity in Şile (1) was so high as to set it quite apart from the rest of the sites.

MFA results illustrate correspondence between variable groups as proximity. In this case, results show that anthropogenic factors correspond more closely with disease severity than do environmental ones (Figure 3a,b). In Figure 3b, the alignment of grafting along the axis most closely associated with severity again indicates its power as a factor.

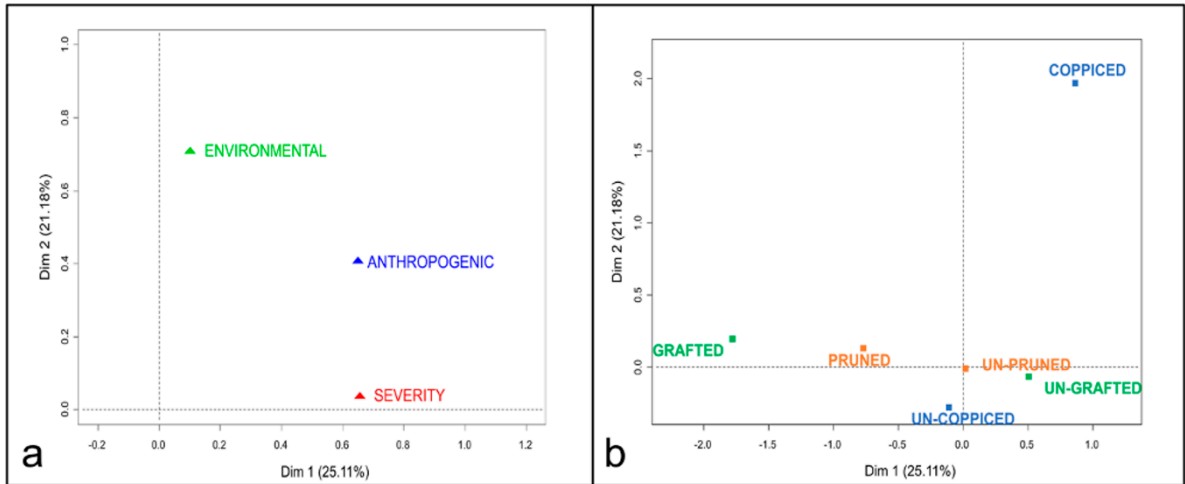

**Figure 3.** (**a**) Results of multiple factor analysis of all factors, with the central position of three factor groups shown; (**b**) Results of multiple factor analysis, with the poles of binary anthropogenic factors shown.

Our next stage of analysis aimed to identify the most meaningful individual factors using analysis by GLMM at the level of trees (*n* = 1500). Our first GLMM produced the analysis shown in Table 2. Figure 4a is a simple visual representation of the relative effects of each factor determined by the model, with significance indicated. The value of the z-score reflects the number of standard deviations away from the mean a given result is, in other words, its significance. We observed a significant positive effect on disease severity by tree diameter and slope, meaning that as tree diameter and slope value increase, so does the likelihood of high disease severity. We see significant negative effects on disease severity by grafting occurrence and environmental wetness (NDVI), meaning that as these variables increase in value, the likelihood of trees having high blight severity decreases.

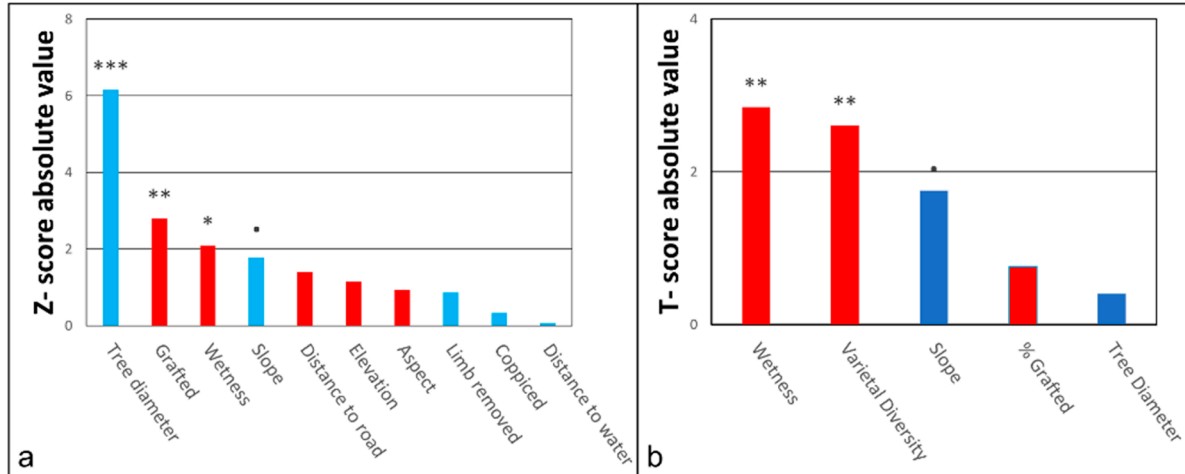

**Figure 4.** (**a**) Effect size of all factors on disease severity for trees as represented by absolute value of z-score generated by GLMM; (**b**) Effect size of significant factors on median plot disease severity as represented by absolute value of t-score generated by LMM. Lighter color indicates a positive correlation and darker, a negative correlation. For both charts, significance indicated as *** representing 0, ** representing 0.001, * representing 0.01, and ● representing 0.05.

Our task was then to determine the effect of the above-listed significant factors in relation to the ethnobotanical knowledge and practices reported by households. This included a Simpson index score for household description of cultivar diversity under management, a Simpson index score for diversity

of plant parts used, and a total number of land access strategies exercised in the collection and use of chestnut materials. As determined by a distinct LMM, of these metrics, only cultivar diversity was determined to have a significant effect on tree health. Our next step thus incorporated ethnobotanical data into a final LMM along with the following biophysical and geographic factors averaged by the number of trees observed in the household-associated plot: tree diameter, environmental wetness, slope, and percentage of grafted trees. Results are shown in Table 3 and Figure 4b. At this level of analysis, we observed significant negative effect on disease severity by environmental wetness and varietal diversity, meaning that as environmental wetness and cultivar diversity increase, the likelihood of trees having high blight severity decreases. We also observed a less pronounced but significant positive effect of slope on disease severity, meaning that as slope increases, so does the likelihood of high disease severity.

**Table 3.** Results of LMM model exploring the effect of significant factors on average plot disease severity as continuous variable. AIC = 216. Factors displayed in order of effect size in Figure 4b. Significance indicated as *** representing 0, ** representing 0.001, * representing 0.01, and ● representing 0.05. DF is degrees of freedom. Pr is probability.

| Factor | DF | Chisq | T-Value | Pr(>Chisq) |
|---|---|---|---|---|
| Wetness | 1 | 8.0928 | −2.845 | 0.00444 ** |
| Varietal Diversity | 1 | 6.7944 | −2.607 | 0.00914 ** |
| Slope | 1 | 3.0764 | 1.754 | 0.79438 ● |
| Percent Grafted | 1 | 0.5760 | −0.759 | 0.447872 |
| Tree Diameter | 1 | 0.1631 | 0.404 | 0.686357 |

Throughout all stages of analysis, the influence of human involvement on the phenomenon of disease severity is clearly shown to be prominent. The nature of this influence is clarified by the mixed effects models which show that grafting and the associated dimension of horticultural knowledge and practice are positively associated with tree health. Considering this prominent positive effect of local livelihood involvement on tree health, we outlined the land access types reported by households who completed the ethnobotanical questionnaire. Reported land access and use codes which households reported exercising are found in Table 4.

**Table 4.** Reported categories of land use or access.

| Land Use or Access Code | Number of Reports |
|---|---|
| Near house on titled land | 33 |
| Near house on untitled land | 9 |
| Away from the house on titled land | 9 |
| On state land under traditional claim | 93 |
| On state land without traditional claim | 2 |
| Treasury, local municipality land | 2 |

Out of 96 households, 93 reported using state land in accordance with locally legitimate claims. Of these 93 households, 27 also reported using the second most common land use or access type, their own property adjacent to the home. Only 3 households reported using or accessing their own deeded land in order to work with chestnut trees. Therefore, it is clear that state forest policy is a significant consideration for our discussion.

## 4. Discussion

Historical, palynological, and genetic evidence suggest that *Castanea sativa* has been sustainably affiliated with anthropogenically maintained space and livelihoods in present-day Turkey for millennia [62,72–74]. Although one very rigorous study suggests a slightly longer timeline regarding

chestnut blight in the east of Turkey [75], it is undoubtedly events and patterns of the last century that have led to the introduction of severe pathogen pressure for the European Chestnut in Turkey. As is common to cases from around the world documented in this Special Issue, this century has marked the introduction of numerous pests and diseases including ink disease by 1951 [7], chestnut blight by 1968 [8], and, more recently, gall wasp by 2014 [9]. This has challenged the viability of rural livelihoods to such an extent that many argue it is the driver of the infamous rural abandonment of certain Black Sea districts [76]. Here, we discuss the factors which we determined to have had a significant effect on tree health to illustrate the importance of continuing human niche construction—embodied in traditional livelihood practices—in the face of rising disease and pest severity. This interpretation is supported by our recurring observation that persistent deliberate human action materializes as the niche, or the material reality experienced by trees (Figure 5). Our interpretation of this history assumes what Johnson and Hunn memorably refer to as "the sophistication of local knowledge of landscape" [77].

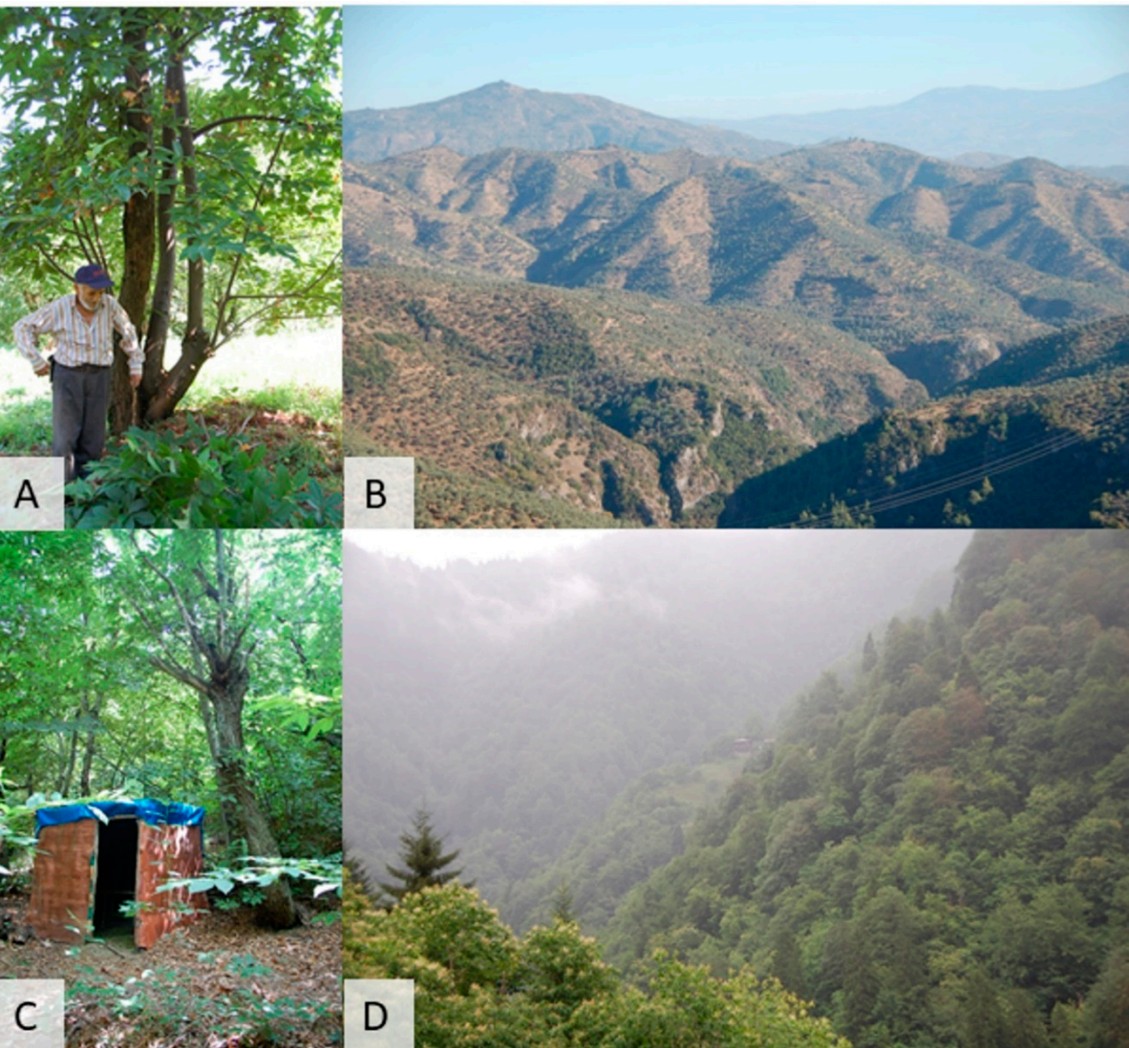

**Figure 5.** Clockwise from top left, (**a**) Bursa province grower in his seventeen-year-old grafted orchard of blight-resistant European-Japanese hybrid and (**b**) new lands of the 49-year leases (kırk dokuz yıllığına kiralık) in Aydın; (**c**) Sayvan, that is, huts found in maintained chestnut groves on state land in Zonguldak province, and (**d**) chestnut inflorescence in the highlands of the eastern Black Sea province of Rize.

The fact that the vast majority of our participants practiced their livelihoods on state land under the direct oversight of the GDF is notable because livelihood management of trees and forest space was observed to be vigorous and comprehensive. In fact, in their totality, our findings illustrate that even plainly biophysical and geographic factors such as tree diameter and the slope occupied by chestnut trees can best be understood as anthropogenically orchestrated. In this sense, specific and concrete practices which affect trees, such as grafting and coppicing, may be secondary to the more powerful ability of local communities to orchestrate, build, and maintain the environment in which trees come to be situated. This is an especially useful observation in our research context as well as others where many routine forest and tree management practices, such as prescribed burning and felling, are strictly prohibited, something which caused interviewees to be reticent in discussing such practices. In the absence of reliable reporting and observation of such practices, broader environmental observations (if understood as a constructed niche) can not only be analyzed for their meta-effect on trees, but they can also be used to make inferences about the underreported practices.

For example, large tree diameter is correlated with high disease severity in the tree-level GLMM. This reflects the reality that more managed groups of trees remain healthier under conditions of disease pressure. Those trees that were most managed were more likely to be culled nearer to the instant when their value in wood was threatened by the advance of pest and disease symptoms. As a second example, the significant factor of slope occupied by chestnut trees is also indicative of management intensity. Especially steep slopes are difficult to work on and are thus least likely to be the sites of tree and orchard maintenance. Where gradual arable slopes are at a premium value, as is the case with tea-producing regions such as Rize, steep inaccessible slopes are commonly left as habitat for the chestnut tree. In totality, our quantitative results indicate that human decision-making is a ubiquitous influence on the physical geographic condition experienced by trees. In other words, the overall niche of chestnut trees is under meticulous and deliberate management in a variety of geographic conditions.

As another example, environmental wetness registers as a prominent factor in the both models, one which might easily be viewed as an environmental factor, but which our observations suggest is both anthropogenic and environmental. The influence of policy is noteworthy in the cases of Aydın and Izmir. An overall dry environment is generally inhospitable for fungal species like *C. parasitica* and *Phytophthora cinnamomi* [78]. The average NDVI value (wetness) for sites in Izmir and Aydın for warm months was 6354 compared to an overall average of 8357. This represents an ecological advantage for the region in terms of chestnut production in the wake of national disease dynamics elsewhere. Participants in these areas recollect that chestnut cultivation was always a feature of the agricultural cycle but has only become preeminent recently, in many cases replacing apple production. This new era of production is evident in the evolution of a unique policy arrangement with the General Directorate of Forestry. These are the 49-year leases (kırk dokuz yıllığına kiralık) that growers in all surveyed villages in the Aegean region have contracted with for the last several years. This represents an important phenomenon for future study, especially when it is considered that successful local community influence of larger bureaucratic structures is an essential prerequisite to successful local community niche construction and biological conservation.

There are specific tree management practices which we directly observed and which we verified to have a significant role in tree health. However, we argue that the demonstrably significant factors of grafting and cultivar diversity are indicative of livelihood approaches that derive more value, both monetary and aesthetic, from the tree species. Our findings on grafting were especially informative and even surprising. Despite being viewed as the quintessentially unnatural intervention [10], grafting has been common with chestnut in the region since at least the Greco-Roman period [72]. As evidenced by the significant genetic divergence of Aegean populations [62], in tandem with continued genetic diversity [79], grafting as a widespread regional practice may be best seen as an indicative factor of a broad landscape-level management strategy for the species. For instance, all growers in the Aegean sites maintained some number of wild specimens (deli/yaban), a fact that drove up their Simpson variety diversity index score.

An individual having a perspective of tree and forest health that is anchored in niche construction can more easily understand the role of a widespread practice like grafting in the context of broad geographic orchestration. The recent history of grafting-dependent, commercial chestnut production in Turkey provides an illustrative example. Historically, chestnut production was dominated by Bursa, which, after major ink disease and chestnut blight epidemics, produces today only 5% of the total national production [10]. This production power has, over time, been positioned in the southern and western areas of Aydın and Izmir, which together produce nearly 48% of Turkey's chestnuts [10]. Growers in Bursa have not remained idle in the face of disease epidemics. What we observed in Bursa was a large and very young generation of trees with the lowest median disease severity of all sites. Trees recorded in Bursa had an average DBH of 11.6 cm compared to the overall mean for all recorded trees of 20.2 cm. Additionally, interviewees in Bursa reported the highest mean number of utilized cultivars per household, 5.5 compared to the group mean of 1.9. Very importantly, Bursa is the only location where a considerable number of "improved" European cultivars was reported. These varieties, such as *Marigoule*, are European-Japanese hybrids selected for their tolerance to chestnut blight. It is clear that growers are enacting an aggressive campaign of experimentation and disease mitigation through horticultural practice. As our study demonstrates, wetter environments like those found in Bursa, despite being hospitable to pests and diseases, are still positively associated with tree health. Over a long enough time frame, the aggressive application of genetic improvements by chestnut-utilizing communities in Bursa may allow these trees to benefit from this natural advantage.

Finally, our findings put a spotlight on the site of Zonguldak, where tree health was uncharacteristically high for the region. In its region, Zonguldak stands out for its notable social characteristics. Alone among Black Sea sites, in Zonguldak we observed the sayvan, a small shed constructed at the site of collection. Here, forest space was maintained much more meticulously. Thorn species (dikenler) and rhododendron (orman gülü) were not observed in the plots. All households reported accessing trees in state forests. No other land access was reported. Yet, this was the one area in the Black Sea where collectors openly claimed to have certain rights to collect in certain areas. Another very interesting feature of Zonguldak situates these findings elegantly. In Turkey, this region is infamous for its coal production. It is likely for this reason that chestnuts from Zonguldak fetch the lowest price in the national markets [59]. Yet, unlike other Black Sea sites which were studied, the coal industry ensured that desirable jobs could be found in these rural spaces. The most evident implication is that many younger families can be found in the villages. These observations paint a rich picture of the rural viability necessary to maintain the niche construction of previous generations. This dynamic dovetails with a rapidly emerging cognizance of the ecological ramifications of rural abandonment [80–82]. We take the case of Zonguldak as a very meaningful one for the Black Sea context, where the chestnut trees are healthy and livelihood practices are vigorous, but where grafting of chestnut trees is categorically unpopular and so plays no role in tree health.

## 5. Conclusions

One of countless challenges to rural viability brought on by economic globalization, the increased movement and severity of exotic diseases and pests, endangers the viability of rural communities worldwide. We document here how the resilience of rural chestnut-livelihood practitioners in Turkey is, in turn, important for the resilience of the chestnut tree population in that country. Policy which supports on-going engagement with the chestnut population by nearby stakeholders is therefore recommended. In Turkey, this would include the continued innovation of the flexible and locally tailored application of forest policy. It is our observation that the plasticity of GDF policy implementation is its greatest strength. The 49-year leases we documented in the southwestern part of Turkey are clearly a local success. Consequently, programs that are similarly derived from local input, and which validate the significant motivations of local communities to maintain the forest, promise similar success. As an example, and given that proper phytosanitary protocol is followed, programs to disseminate locally appreciated and demonstrated vigorous germplasm, for grafting and/or direct planting, can

augment the natural motivations of local livelihood practitioners. In addition, legally enshrining the construction and use of harvest huts, or sayvanlar, across the Black Sea Region would stoke popular motivations to maintain forest areas. These huts do not just meet the pragmatic needs of harvest labor and post-harvest management, they are a beloved site of forest leisure, enjoyment, and memory.

Our findings suggest that, in the face of rising pest and disease severity, human engagement with chestnut trees and forest space increases tree health. More specifically, the liberty to conduct traditionally obligatory tree and landscape management was positively related to healthier chestnut tree populations. We assume that the identification of clear indicators of heightened human engagement with and interest in individual trees such as our observations of grafting and culling is a promising approach to developing policies which may encourage human niche construction. Precise knowledge of the most important, feasible, and appropriate anthropogenic contributions to the tree species can support state conservation efforts and local livelihoods. Regarding grafting and cultivar diversity, these anthropogenic factors are a significant correlate of tree health. However, the higher tree health of Zonguldak suggests that the freedom of local chestnut-utilizing communities to access chestnut forest space and perform the maintenance inherent to traditional livelihoods underpins tree health as well. Consequently, our research highlights the power of socially targeted and locally informed policies to reinforce vigorous and motivated human niche maintenance in forests and other rural geographies.

**Author Contributions:** All field research was conducted by J.W., C.K., N.K., and T.O. All data analysis and manuscript conceptualization was undertaken by J.W., C.K., N.K., E.B.A., and T.O., E.B.A. and T.O. conducted the literature review of relevant official forest policy documents. D.J. facilitated and oversaw statistical analysis and theoretical strategy. S.A. provided continual advice regarding manuscript production and research design.

**Funding:** This work was supported by the American Research Institute in Turkey, 3260 South Street, Philadelphia PA 19104, and the U.S. Borlaug Fellows in Global Food Security Program, Center for Food Security, Purdue University (Sponsor Agreement ID 8000079396).

**Acknowledgments:** The authors are highly indebted to the Turkish National General Directorate of Forestry, especially their offices and personnel in Borçka, Bursa, Çanakkale, Rize, Şile, Sinop, Trabzon, and Zonguldak, for their invaluable advice and support. They are also very thankful to the administration of Istanbul University-Cerrahpasa Faculty of Forestry for facilitating their research efforts. The authors wish to thank Yilmaz Erdal, chair of the Anthropology Department at Hacettepe University, for his invaluable advice. They sincerely thank the Scientific and Technological Research Council of Turkey (TUBITAK), the American Research Institute in Turkey, the U.S. Borlaug Fellows in Global Food Security Program, and the Turkish Fulbright Commission for their generous support of this research. Finally, the authors are extremely grateful for the extensive assistance of the Cornell Statistical Consulting Unit, especially Lynn Johnson.

**Conflicts of Interest:** The authors declare no conflict of interest.

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
