# Peer review of "The Role of Traditional Livelihood Practices and Local Ethnobotanical Knowledge in Mitigating Chestnut Disease and Pest Severity in Turkey"

_forests, doi:10.3390/f10070571_

Round 1
Reviewer 1 Report
This research article is very interested and have value for publishing in Forests.
Line 52:
Can you elaborate the meaning ”the center of diversity” more clearly?
Introduction
It is better to state your research objectives in the Introduction parts more clearly.
Line 154-155
In Easter Black Sea side, what kind of operation did either people or government took to the hard wood forests.
Line 167-168
In each site, the authors selected 8-10 household for your interview. Please describe what king of criteria you did used for selection of households ? For example, household who use (or collect) chestnut regularly or others?
Line 243-244
96 households participated your interview, but you have 97plot. Dose this means one household did not participate interview or one household have two plots or others?
Line 261- 263
For good structure of article, the location of Figure2 and table1 should be changed.
Line348-349
Are there any evidence the pattern of last century?
Line 438-439
Do you have some concrete idea for providing forest-related policies?
Line 519-520
The fonts of journal title should be changed italic.
Line 547
The word of “a” between Briscoe, D. and Frankham, R. in the author name may be deleted.
Line 558-559
No journal name
Line 560-562 It may be wrong journal information.
Line 570-571 The publish year of journal article may be wrong.
Line583-584
The volume number of the article is missing, instead of “December”
Line589-590 The journal name, volume number and page number of the article is missing.
Author Response
Thank you so kindly for your thorough review. We have attempted to make a thorough accounting of all comments. Please see attached document for an item by item description of our responses.

Reviewer 2 Report
line 58 consider adding citation on Cryphonectria parasitica from Serbia, last information.
Karadžić, D., Radulović, Z., Sikora, K., Stanivuković, Z., Golubović Ćurguz, V., Oszako, T., & Milenković, I. (2019). Characterisation and pathogenicity of Cryphonectria parasitica on sweet chestnut and sessile oak trees in Serbia. Plant Protect. Sci, 55, 191-201.
line 68 "are" is in italic and line 82 "maintainance" as well, was it by purpose?
line 122 what do you mean by "pg 7" or line 123 by "pp 18"?
line 136 "262.000" instead of coma maybe dot or nothing?
line 143 "masl" appears for the first time maybe explain this abrevaition?
Author Response

(The authors gave the same response as above.)
